# Protective Effect of the Naringin–Chitooligosaccharide Complex on Lipopolysaccharide-Induced Systematic Inflammatory Response Syndrome Model in Mice

**DOI:** 10.3390/foods13040576

**Published:** 2024-02-14

**Authors:** Sheng Tang, Zhu Ouyang, Xiang Tan, Xin Liu, Junying Bai, Hua Wang, Linhua Huang

**Affiliations:** 1Citrus Research Institute, Southwest University, Chongqing 400700, China; t2353749207@163.com (S.T.); 15213108545@163.com (Z.O.); xiangtan@swu.edu.cn (X.T.); lxqq1985734352@email.swu.edu.cn (X.L.); baijunying@swu.edu.cn (J.B.); wanghua@cric.cn (H.W.); 2National Citrus Engineering Research Center, Chongqing 400700, China

**Keywords:** naringin, chitooligosaccharides, complex, systemic inflammation, lipopolysaccharide

## Abstract

Naringin is one of the common flavonoids in grapefruit, which has anti-cancer, antioxidant, and anti-inflammatory activities. However, its poor solubility limits its wide application. Therefore, the aim of this study is to investigate the anti-inflammatory effect of naringin combined with chitooligosaccharides with good biocompatibility by constructing a mouse model of systemic inflammatory response syndrome (SIRS). The results showed that the naringin–chitooligosaccharide (NG-COS) complex significantly inhibited lipopolysaccharide (LPS)-induced weight loss, reduced food intake, tissue inflammatory infiltration, and proinflammatory cytokines IL-6, TNF-α, INF-γ, and IL-1β levels. The complex also significantly affected the content of malondialdehyde and the activities of MPO, SOD, and GSH in the liver, spleen, lungs, and serum of mice with systemic inflammation. In addition, NG-COS significantly inhibited the mRNA expression of inflammatory factors in the TLR4/NF-κB signaling pathway. Principal component analysis showed that the complexes could inhibit LPS-induced systemic inflammation in mice, and the effect was significantly better than that of naringin and chitooligosaccharides alone. This study explored the synergistic effects of chitosan and naringin in reducing inflammation and could contribute to the development of novel biomedical interventions.

## 1. Introduction

Naringin is one of the most common flavonoids in citrus peel, which is composed of glucose, rhamnose, and naringenin [1]. Because of its antioxidant [2], antitumor [3], anti-inflammatory [4], antibacterial, and other biological activities [5], naringin has potential preventive and therapeutic effects on human diseases. For example, naringin is thought to enhance the function of the hepatic antioxidant system as well as the metabolism of hepatotoxic substances [6], improve insulin resistance and glucose uptake by inhibiting enzymes that metabolize carbohydrates [7]. In addition, naringin alleviated LPS-induced intestinal barrier damage by inhibiting inflammatory factors, improving antioxidant function and intestinal tight junctions [8,9]. However, due to its low water solubility and strong bitterness, it is rarely used directly by absorption and metabolism [10]. It is particularly important to improve the processing availability of naringin. By combining with other substances to form complexes, naringin may be better absorbed and utilized to achieve better effects. It has been demonstrated that the naringin–copper complex exhibits higher anti-inflammatory activity than free naringin [11]. The cross-linking of naringin and chitosan enhances its potential as a composite film to inhibit lipid oxidation in foods [12] Therefore, the preparation of naringin complexes is an effective method to improve its biological activity.

Chitooligosaccharides (COSs) are sugar chains composed of 2 to 20 glucosamines linked by β-1,4 glycosidylinkages and form a component of chitosan or chitin. Studies have shown that COSs manifest anti-inflammatory activity and antioxidant action via NF-κB, Erk1/2, Akt and Nrf2/HO-1 signaling [13]. In addition, they have good biocompatibility and biodegradability, so are often used as a complex construction carrier to enhance the solubility of the matrix itself. For example, COSs could serve as a carrier for β-carotene delivery. The complex did not cause the loss of the radical scavenging activity of β-carotene and guaranteed its water solubility, and can be absorbed through the intestine, thus quickly entering the blood, and producing systemic biological effects on the body. Studies have shown that COSs conjugate effectively to reduce acute kidney injury in mice compared with free COSs [14]. In addition, the molecular weight of COSs is highly correlated with their biological activities [15,16], which are essential for their application and utilization. A high molecular weight may improve the effectiveness of COSs during drug delivery while mitigating potential toxicity or adverse effects. Low-molecular-weight COSs usually have better solubility and are convenient for use in various applications.

Systemic inflammatory response syndrome (SIRS) often occurs along with harmful stressors, such as infection, trauma, surgery, acute inflammation, ischemia, reperfusion, or malignant tumors [17]. SIRS is a physiological response in which the body secretes cytokines and inflammatory mediators in response to external stimuli. SIRS can be triggered by exogenous macromolecules such as endotoxin [18], also known as lipopolysaccharide (LPS), acting on Toll-like receptors (TLR). In the process of SIRS, excessive release of proinflammatory factors causes inflammatory cascade reactions, which eventually leads to leukocytosis, decreased immunity, and metabolic dysfunction. Therefore, inhibiting the overproduction of chronic proinflammatory cytokines is important for the development of anti-inflammatory therapies aimed at preventing and alleviating inflammatory diseases. COSs and naringin each have certain biological activities [15], and they may enhance each other’s activities through interaction or synergistic effects in some cases. Complexes may exhibit stronger anti-inflammatory effects; however, it should be pointed out that the specific effect depends on the ratio of the complex, the preparation method, and the specific study conditions applied.

In our previous study, we have prepared a naringin–chitooligosaccharide (NG-COS) complex, but the synergistic anti-inflammatory effect of naringin and COS was not investigated. Based on this, in the present study, LPS was used as the initiating factor of the inflammatory cascade to induce systemic inflammation in mice. To investigate the anti-inflammatory effects of naringin, COSs of different molecular weights and their effect on systemic inflammation in mice via intragastric administration of naringin, COSs, and their complexes were investigated. To further understand the potential mechanisms of naringin, COSs, and their complexes, we also tested their effects on body weight, organ index, inflammatory markers, and intestinal structural pathology in mice. Finally, through the correlation analysis, this paper discusses gene expression of TLR/NF-κB signaling pathways and naringin complex relationships with drug delivery. Our study will provide a theoretical foundation for the research and development of naringin in medical care, dietary supplements, and livestock and poultry feed, and provide theoretical reference for the comprehensive utilization of citrus peel.

## 2. Materials and Methods

### 2.1. Animal Experiments

Eighty female and male Kunming mice (20–24 g, SPF) were obtained from Hunan Slake Jingda Experimental Animal Co., LTD (Hunan, China). Mice were adaptively fed at a temperature of 22 ± 4 °C and a humidity of 50% ± 20%. After one week of adaptive feeding, all mice were randomly divided into eight groups (*n* = 10 per group, half male and half female mice): (1) The control (CT) group: gavage daily with 200 μL of sterile normal saline for 2 weeks. (2) The LPS (LPS) group: gavage daily with 200 μL of sterile normal saline for 2 weeks. (3) The dexamethasone (DXMS) group: gavage daily with dexamethasone for 2 weeks (100 mg/kg of body weight). (4) The naringin group: gavage daily with naringin for 2 weeks (100 mg/kg of body weight). (5) The chitooligosaccharide A (COSA) group: gavage daily with chitooligosaccharide A (molecular weight 2000–3000 Da, Shanghai Yuangye Biotechnology Co., LTD (Shanghai, China)) for 2 weeks (100 mg/kg of body weight). (6) The chitooligosaccharide B (COSB) group: gavage daily with chitooligosaccharide B (molecular weight 800–1000 Da, Shanghai Yuangye Biotechnology Co., LTD (Shanghai, China)) for 2 weeks (100 mg/kg of body weight). (7) The naringin–chitooligosaccharide A complex (NG-COSA) group: gavage daily with naringin–chitooligosaccharide A complex (naringin to chitooligosaccharide A = 1:5) for 2 weeks (100 mg/kg of body weight). (8) The naringin–chitooligosaccharide B complex (NG-COSB) group: gavage daily with naringin–chitooligosaccharide B complex (naringin to chitooligosaccharide B = 1:5) for 2 weeks (100 mg/kg of body weight). During the experiment, the mice were fed and watered AD libitum. After 14 days, the mice in the CT group were intraperitoneally injected with normal saline, and the mice in the other groups were intraperitoneally injected with 6 mg/kg of LPS (O111:B4, Sigma Aldrich, St. Louis, MO, USA) to induce inflammation, and daily gavage was continued (Figure 1A) [19,20,21]. The body weight and food intake of the mice were measured at 10:00 every day during the period of intervention and lipopolysaccharide-induced inflammation. After 4 days of LPS-induced inflammation, the mice were fasted for 24 h, and after 6 h of LPS-induced inflammation on the 5th day, the mice were dissected after eyeball blood collection.

After the experiment, all mice were anesthetized by intraperitoneal injection of 1% pentobarbital sodium (dose of 45 mg/kg) and sacrificed for cervical dislocation. The blood samples of mice were collected and centrifuged at 3000× *g* for 30 min to collect the serum for further analysis. The liver, kidney, spleen, lung, and heart organs were carefully harvested, and the organ index was calculated as a percentage of the body weight of the mice.

### 2.2. Histology Examination

After the animal experiment, the liver, spleen, and lung tissues of the mice were collected and immediately put into a solution containing 4% paraformaldehyde for fixation. The fixed liver, spleen, and lung tissue was dehydrated with gradient ethanol solution. The dehydrated liver tissue was permeated with xylene for 20 min and then immersed in wax for 2 h. Finally, the tissue was embedded in paraffin with a melting point of 52–56 °C, and the embedded tissue was placed on the freezing table within 10 min to quickly cool and the wax block was removed. The wax blocks were sectioned using a microtome, and the sections were put into a warm water bath at 48 °C. The adhesive slides were used to catch the pieces in the water, and the pieces were burned at 50 °C for more than 30 min. The sections were deparaffinized in xylene, followed by hydration in absolute ethanol aqueous solution. The hydrated tissue sections were stained with hematoxylin-eosin (H&E) according to the instructions of the HE staining kit and observed under a light microscope. The pathological section description of tissues was referred to the previous report [22].

### 2.3. Inflammatory Cytokines Examination

After the animal experiment, 0.1 g of liver, spleen, and lungs tissue was collected and stored in liquid nitrogen for the extraction of the total protein [23]. Ice-cooled phosphate-buffered saline (PBS) (1:9 *w*/*v*) was added to the mechanical homogenization, and then centrifuged (4 °C, 3000 rpm for 20 min) to collect the supernatant. The contents of interferon gamma (IFN-γ), interleukin-6 (IL-6), tumor necrosis factor-α (TNF-α), interleukin-17 (IL-10), and interleukin-1β (IL-1β) in the serum, liver, spleen, and lungs tissues were determined by using an enzyme-linked immunosorbent assay (ELISA) kit (R&D System China Co., Ltd, Shanghai, China) according to the manufacturer’s instructions. The analysis methods were carried out according to the manufacturer’s instructions.

### 2.4. Myeloperoxidase (MPO) Activity and Oxidative Stress Indicator Examination

Appropriate amounts of liver, spleen, and lung tissues were added to PBS solution at a weight-to-volume ratio of 1:9, and the tissues were homogenized thoroughly using a homogenizer. The samples were centrifuged at 3000 rpm (4 °C) for 20 min, and the supernatant was fractionated for later use. Myeloperoxidase (MPO) and oxidative stress markers including superoxide dismutase (SOD), malondialdehyde (MDA), and glutathione (GSH) in the serum and liver, spleen and lung tissues were measured by ELISA according to the instructions of the kit (R&D System China Co., Ltd, Shanghai, China). The analysis methods were carried out according to the manufacturer’s instructions.

### 2.5. Quantitative Real-Time Polymerase Chain Reaction (qRT-PCR) Analysis

The total RNA was extracted from the liver, spleen, and lungs tissue with Trizol reagent (Invitrogen, Carlsbad, CA, USA) and was quantified using NanoDrop. Further, the RNA extract was reversely transcripted into cDNA by utilizing RevertAid First Strand cDNA Synthesis Kit (Thermo Fisher Scientific, Waltham, MA, USA). A 20 μL reverse transcription reaction mixture consisted of RNA (1 μL), primer dT (1 μL), water (10 μL), Ribolock RNase inhibitor (1 μL), reaction buffer (4 μL), Revert Aid M-muLV (1 μL), and 10 mM dNTP mix (2 μL). The cDNA was mixed with the SYBR Green PCR Master Mix (Thermo Fisher Scientific, Waltham, MA, USA) and was used for relative quantification of the mRNA expression level of various genes in the TLR/NF-κB pathway such as Toll-like receptor 4 (TLR4), myeloid differentiation factor (MyD88), IL-6, TNF-α, IL-1β, IκBa, and NF-κB p65. RT-PCR was conducted on a 7500 Real-time PCR System (Applied Biosystems, Foster city, CA, USA). The cycle program was set to 95 °C for 10 min to 95 °C for 15 s, and to 60 °C for 60 s for 40 cycles. The semiquantitative calculations were carried out according to formula 2^−ΔΔCt^ using β-actin as an internal reference. The primer sequence sets used are listed in Appendix A.

### 2.6. Statistical Analysis

Data collation and analysis was carried out and charts were made using Microsoft Office Excel 2019, SPSS 26.0, and GraphPad Prism 8.0.2. The significant difference between the groups was calculated by one-way ANOVA with Fisher’s LSD post hoc test. The results between different groups were considered statistically significant when *p* < 0.05. Principal component analysis (PCA) combined with a heatmap was implemented to visualize the difference between groups. OmicStudio tools (https://www.omicstudio.cn/tool, accessed on 21 November 2021) was used to parallelize the data of each group and then perform PCA dimensionality reduction visualization.

## 3. Results

### 3.1. Effect of the Complex on Body Posture, Body Weight, and Food Intake in LPS-Induced SIRS Model Mice

Systemic inflammation can cause headaches, fatigue, loss of appetite, and swelling and pain in the anus. As can be seen from Figure 1B, during prophylactic gavage, compared with the normal group, the food intake of mice in the other administration groups increased significantly, among which the body weight of mice increased most significantly in the naringin and NG-COSB groups (Appendix A), indicating the effect of naringin and NG-COSB in terms of increasing appetite and promoting growth. During LPS-induced inflammation, the body weight and food intake of mice in the LPS group were significantly lower than those in the CT group (Figure 1C,D). However, DXMS intervention significantly increased the body weight and food intake of LPS mice, while the body weight and food intake of mice in the naringin group, NG-COSA group, and NG-COSB group exhibited no significant changes. After LPS administration, compared with the normal group, the perianal yellow-–brown foreign bodies increased, with foreign body adhesion, the mice moved slowly, the body shivered, the hair became unsmooth, and there were secretions in the corner of the eye (Figure 1E). The perianal foreign bodies in the naringin group, the COSA group, and the COSB group were reduced, but the mental state of the mice was still atrophied. The perianal foreign bodies in the NG-COSA group, the NG-COSB group, and the DXMS group were significantly reduced, and the mental state of the mice was similar to that of the CT group. Compared with naringin, NG-COSB had a better intervention effect on LPS mice, but the difference was not statistically significant.

These results indicate that LPS causes weight loss and loss of appetite in mice and that DXMS intervention inhibits LPS-induced weight loss and anal foreign bodies. NG-COSA and NG-COSB could alleviate LPS-induced systemic inflammation, while there was no significant difference with NG, COSA, and COSB intervention.

### 3.2. Effect of the Complex on the LPS-Induced Viscera Index of the Mouse Model of SIRS

Inflammation is typically characterized by inflammatory cell infiltration and tissue edema. As shown in Figure 2B, LPS could significantly increase the organ indexes of the kidney, heart, spleen, liver, and lung in mice, indicating that LPS caused tissue edema or other pathological changes in mice. Naringin, NG-COSA, and NG-COSB can inhibit the changes in the heart, spleen, and lung tissues caused by LPS. DXMS can significantly reduce the organ index of the kidney, heart, lung, and spleen, and reduce the tissue edema caused by LPS. This indicates that naringin, NG-COSA, NG-COSB, and DXMS can effectively reduce the organ index. DXMS, NG-COSA, and NG-COSB increased the organ index of the liver in mice, which may be due to the increased food intake of mice and their effect on liver metabolism. Naringin, NG-COSA, NG-COSB, and DXMS all affect the function of the spleen, which indicated that they can possibly affect the immune function of the body.

The spleen, liver, and lung tissues of the mice were stained with H&E, and the histopathological changes were observed under a light microscope, as shown in Figure 2A. The structure of the spleen tissue in the CT group and the DXMS group was tight, the shape was regular, the layers, small structure, and the white pulp and red pulp of the spleen were obvious. The spleen tissue structure of mice in the LPS group was loose, the outline of the small structure of the spleen was fuzzy, the shape was irregular, and the neutrophil infiltration was obvious. Compared with the LPS group, the spleen tissue structure of the CT group was compact, the layers were obvious, and the neutrophil infiltration was significantly reduced. The spleen of mice in the NG group, the COSA group, and the COSB group had a loose structure, irregular morphology, and reduced neutrophil infiltration. The spleen tissue structure of mice in the DXMS group, the NG-COSA group, and the NG-COSB group was tight and regular, the number of neutrophils was significantly reduced, and the degree of inflammation was significantly reduced.

The liver and lung tissue structure of the CT group was normal, there was no inflammatory cell infiltration. After LPS treatment, the mice’s liver tissue structure was disordered, the liver cells were swollen, and a large number of inflammatory cells had infiltrated the organ; the lung tissue structure of the mice was changed, the alveolar wall was thickened, a large number of inflammatory cells had infiltrated. Compared with the LPS group, there was a large inflammatory cell infiltration in the liver tissue of the COSA group and the COSB group, and the inflammatory cell infiltration was lower and the liver tissue structure was disordered in the naringin group. The naringin, COSA, and COSB groups also had significant inflammatory cell infiltration and alveolar wall thickening. However, preconditioning with NG-COSA and NG-COSB significantly reduced LPS-induced liver and lung tissue damage. These results suggested that NG-COSA and NG-COSB could reduce the degree of pulmonary inflammation in LPS-induced systemic inflammation. In general, naringin, COSA, COSB, NG-COSA, and NG-COSB showed a certain inhibitory effect on the SIRS status in mice with inflammation.

### 3.3. Complex Regulate Inflammatory Cytokine Levels

The effect of naringin, COSA, COSB, NG-COSA, and NG-COSB on inflammatory status was examined and is shown in Figure 3. As shown in Figure 3A–D, LPS treatment significantly promoted the content of pro-inflammatory factor INF-γ, TNF-α, IL-6, and IL-1β in the serum, spleen, liver, and lung tissues of the mice. By contrast naringin, COSA, COSB, NG-COSA, and NG-COSB administration obviously reduced the levels of pro-inflammatory factors IL-6, TNF-α, INF-γ, and IL-1β in the spleen, liver, and lung tissues of the mice. Compared with the normal group, there was no significant difference in anti-inflammatory factor IL-10 in the model group, while naringin, NG-COSA, and NG-COSB intervention significantly increased the release of IL-10 (Figure 3E). Notably, the inhibitory effect of NG-COSA and NG-COSB intervention on pro-inflammatory factors in the lung was significantly stronger than that of the naringin and COS (A, B) groups. The promotion effects of NG-COSA and NG-COSB on IL-10 were significantly stronger than those of COSA and COSB. In conclusion, these results indicate that NG-COSA and NG-COSB have a certain inhibitory effect on the inflammatory state of tissues in mice with systemic inflammation.

### 3.4. Complex-Reduced MPO Activity and Oxidative Stress Indicators

Neutrophils are the most abundant white blood cells in humans, and MPO is their most expressed protein, often detectable in injured tissues. LPS induction increased MPO content in the serum and liver, spleen, and lung tissues of mice. Compared with the LPS model group, the MPO content of serum, spleen, liver, and lung tissues was significantly decreased by 27.8%, 25.6%, 29.2%, and 26.8% (*p* < 0.05) in the NA group and 27.8%, 19.7%, 33.5%, and 23.5% (*p* < 0.05) in the NG-COSB group. The NG-COSA group and the NG-COSB group had lower MPO levels in the serum and tissues than the LPS group, and there were significant differences in the MPO of the lung and spleen tissues between the two groups (*p* < 0.05) (Figure 4A).

There is a mutually promoting relationship between oxidative stress and inflammatory response. SOD and GSH are important antioxidant factors that help the body to eliminate oxidative stress. MDA is an important product of lipid peroxidation and its content reflects the degree of peroxidation in the body. Compared with the CT group, the activities of SOD and GSH in serum, spleen, lung, and liver of the LPS group were significantly decreased by LPS. Compared with the LPS group, the NG-COSA and NG-COSB groups had significant increases in SOD and GSH content in the serum, spleen, liver, and lung tissues (*p* < 0.05) (Figure 4B,C). The levels of SOD in the serum and the tissues of mice in the NG-COSB group and the NG-COSA group were higher than the LPS group, and there was no significant difference between the two groups (*p* > 0.05). Compared with the CT group, the MDA content in the serum, spleen, lung, and liver of the LPS group increased significantly, and the MDA content in the serum, spleen, liver, and lung tissues of other the intervention groups decreased significantly (Figure 4D). To sum up, LPS caused increased oxidative stress and decreased antioxidant capacity in the tissues and serum. However, naringin, COSA, COSB, NG-COSA, and NG-COSB all ameliorated LPS-induced oxidative stress, and NG-COSA and NG-COSB intervention were significantly better than COS intervention alone.

### 3.5. The Complex Inhibited the mRNA Expression of Inflammatory Cytokines

The results show that the mRNA-relative expression levels of TLR4, MyD88, NF-κB p65, IκBa, TNF-α, IL-1β, and IL-6 mRNA in the liver, lung, and spleen tissues of mice in the LPS group were significantly higher than those in the CT group (*p* < 0.05) (Table 1, Table 2 and Table 3). It was confirmed that LPS could upregulate a series of genes in the TLR/nf-kb pathway to induce an inflammatory response. The relative expression levels of the liver, lung, and spleen tissue in the naringin group, the COSA group, the COSB group, the NG-COSA group, and the NG-COSB group were significantly lower than those in the LPS group (*p* < 0.05), and the relative expression levels in the NG-COSA group and the NG-COSB group were closer to those in the CT group and the DXMS group. The secretion of TNF-α, IL-1β, and IL-6 in the liver, lung, and spleen tissue was reduced by downregulating the gene expression of these inflammatory cytokines. These results suggest that NG-COSA and NG-COSB can possibly inhibit the activation of the TLR4/NF-κB signaling pathway. It also provides further evidence that NG-COS is protective against LPS-induced systemic inflammatory syndrome in mice.

### 3.6. The Inhibition of SIRS Is Related to TLR4/NF-κB Signaling Pathways and Oxidative Stress

In order to further analyze and compare the protective effects of naringin, COSA, COSB, NG-COSA, and NG-COSB on SIRS mice, different groups of experimental mice were used for visualization analysis. Principal component analysis (PCA) combined with a heatmap was used to visualize the content of inflammatory factors and oxidative stress factors in the serum and liver, spleen, and lung tissues (Figure 5A,B), as well as the relative expression of inflammatory factors in the liver, spleen, and lung tissues (Figure 5C). The results showed that inflammation was obvious in the LPS group, but not in the CT group. The DXMS group, naringin group, COSA group, COSB group, NG-COSA group, and NG-COSB group could inhibit the expression and release of inflammatory factors, increase the contents of SOD and GSH, and reduce the contents of MDA and MPO. At the same time, in the visualization results, we can also see that the intervention effects of the same dose of COS and naringin are basically the same. Compared with naringin, COSA has a stronger antioxidant activity, rather than reducing the release of inflammatory factors. The overall inhibitory effect on SIRS was DXMS > NG-COSB > NG-COSA > naringin > COSB > COSA.

## 4. Discussion

Advances in the study of flavonoids have been fascinating, but their low bioavailability is of concern. It has been found that a cell model, Caco-2 cells, reveals that hydrophobic flavonoids are better absorbed due to their higher permeability to the phospholipid bilayer of the cell membrane. Naringin, as the main flavonoid compound in citrus, affects its digestion and absorption in the body because of its connection with glycosides. Chitosan is a natural and abundant biopolypolymer that is deacetylated from chitin and is widely used due to its biodegradability and biocompatibility [24,25]. Many studies have reported the use of chitosan as an effective strategy to further improve the efficiency of hydrophobic drug loading [26]. COS, compared with chitosan, has a smaller molecular weight and more biological activity. It is usually composed of 2 to 20 glucosamine residues. Small molecular COS, between a few hundred and a few thousand Da, may have some advantages in biological activity, immune regulation, etc. In the present study, Kunming mice were selected as the experimental subjects, and LPS was injected intraperitoneally to induce inflammation to construct a mouse systemic inflammation model. In this study, male and female evenly mixed mice were used to explore the effects of the complex on inflammation in mice of different genders, and to ensure the broad applicability of the experimental results. The menstrual cycle of the mice was predicted by observing the external genitalia of the female mice and recording the changes in body weight to reduce the influence on the experimental results. To compare the anti-inflammatory activities of naringin, COS, and NG-COS in vivo. The various phenotypic data show that LPS-induced inflammation was successful and the body weight of and amount of food consumed by the mice decreased. In addition, NG-COS-treated mice showed increased food intake during prophylactic gavage, possibly because both naringin and COS could regulate glucose and lipid metabolism to promote appetite [27,28]. H&E staining showed that intraperitoneal injection of LPS caused inflammation in the liver, spleen, and lung tissues of the LPS group, with obvious inflammatory cell infiltration. The inflammatory cell infiltration was weakened in the DXMS group, the NG-COSA group, and the NG-COSB group, and the morphology of tissue cells was normal. However, the intervention effects of the naringin and COS groups were weaker than positive drugs (the DXMS group) and NG-COS. The complexes may produce synergistic effects that somehow enhance anti-inflammatory properties or improve their stability in the organism [29]. We firstly evaluated the effects of naringin and COS on the levels of serum, the tissue inflammatory factors, and oxidative stress factors, and found that the intervention with naringin, COS, and NG-COS reduced the levels of INF-γ, TNF-α, IL-6, and IL-1β in the serum and tissues, and increased the production of anti-inflammatory factors and antioxidant factors. To explore the mechanism of action of naringin and COS, the TLR4/NF-κB signaling pathway was detected in the liver, spleen, and lung tissues of the mice.

TLR4 is a member of the TLR family, which can recognize microbial-associated molecular patterns (MAMP), such as LPS, and participates in the regulation of innate and adaptive immunity. When LPS stimulates the expression of TLR4, it induces a cascade of activation of NF-κB and other signaling [30,31]. Once activated, NF-κB can regulate the expression of inflammatory genes and the release of cytokines, such as TNF-α, IL-1β and IL-6, and aggravate inflammatory damage. Subsequently, we measured the inflammatory response to LPS-induced liver, lung, and spleen tissue and found that the intervention with naringin, COS, and NG-COS significantly reduced the expression of pro-inflammatory cytokines TNF-α, IL-1β, and IL-6. NG-COSA and NG-COSB could significantly downregulate the mRNA-relative expression of TLR4, NF-κB p65, IκBα, and MyD88 in the TLR4/NF-κB pathway. In this study, the protein level of the TLR4/NF-κB pathway was not further studied and verified. The process of the inflammation induced by LPS can produce ROS, which can not only directly start the NF-κB pathway but can also lead to oxidative stress [32]. MPO is a pro-oxidative and pro-inflammatory peroxidase secreted by neutrophils. MPO can induce lung injury and is related to the pathogenesis of the kidney [33,34]. When LPS acts on mice to cause systemic inflammatory syndrome, MPO will enter the extracellular fluid with the infiltration of neutrophils, resulting in increased MPO content in the tissues, thereby aggravating the inflammatory response. SOD is an important antioxidant enzyme in the body and plays an important role in the balance of oxidation and antioxidation. SOD is a part of the enzymatic defense system that resists oxidative decay by converting superoxide radical anion to H_2_O_2_ [35] The increase in SOD activity increases the resistance to oxidative stress. Surprisingly, NG-COSA and NG-COSB can inhibit the excessive oxidative stress response induced by LPS. They significantly increased the contents of SOD and GSH, decreased the content of MDA, and decreased the content of MPO in the tissues and serum. PCA analysis and heatmap visualization finally revealed a correlation between the suppression of inflammation and the NG-COS intervention. The PCA and heatmap results showed that the NG-COSA group and the NG-COSB group were closer to the CT group and the DXMS group, and there was no significant difference between the two groups. PCA analysis and heatmaps also showed that NG-COS intervention inhibited LPS-induced systemic inflammation significantly better than naringin and COS alone. In addition, visualization analysis showed that COSA, which has a larger molecular weight, seemed to have a stronger antioxidant effect than naringin and COSB.

Interestingly, COS, with its much stronger bioavailability than naringin, showed similar anti-inflammatory effects with naringin at the same dose. However, the combined complex, after reducing the naringin dose, was found to have a significantly stronger effect on mice with systemic inflammation than either of them alone in our study. Studies have reported that COS can also affect intestinal flora to improve colitis and regulate lipid metabolism. COS may have a certain synergistic anti-inflammatory effect with naringin by regulating the composition of intestinal microbiota and short-chain fatty acids [36]. Perhaps exploring its mechanism of action from multiple perspectives, such as intestinal flora and metabolomics, will lead to different findings. Previous studies have reported that COS can reconstruct the intestinal mucus structure related to drug absorption, including thickness, pore space, viscosity, etc. This proabsorptive strategy, in particular, enhances the oral absorption of water-soluble drugs, potentially reducing toxicity and improving efficacy [37]. Studies have shown that the water solubility of naringin is significantly improved and the bitter taste of naringin is effectively reduced by benefiting from the interaction between naringin and COS [38]. COS may help to increase the stability and solubility of naringin and thereby increase its bioavailability. In addition, NG-COS enhanced antioxidant capacity and antibacterial properties [38]. Indeed, the complexes of COS with different degrees of polymerization and naringin have similar effects, and their anti-inflammatory activities are better and closer to DXMS. However, COS with a single degree of polymerization is still a mixture of many uncertain sequences and it is difficult to determine which structure is responsible for biological effects [39]. Therefore, the physiological function and the structure–activity relationship of COS are worthy of further study. Our study found that COS-NG could be anti-inflammatory by regulating inflammatory pathway genes and antioxidant activity, and suggested that COS could improve its anti-inflammatory activity by increasing the bioavailability of naringin.

## 5. Conclusions

In conclusion, NG-COS has better protective effect on LPS-induced systemic inflammation in mice compared to naringin and COS, and its mechanism may be related to the regulation of the TLR4/NF-κB signaling pathway to inhibit inflammatory response and improve oxidative stress injury. These better biophysical properties obtained by NG-COS complexation suggest that the anti-inflammatory effect may be enhanced by improving the bioavailability of naringin, which validates the great advantages of functional polyphenols interacting with polysaccharides, thereby improving the quality of natural products and broadening their applications in food.

## 6. Patents

Patents resulting from the work reported in this manuscript are: ZL 2022 1 0692080.6 and ZL202110352150.9.

## Figures and Tables

**Figure 1 foods-13-00576-f001:**
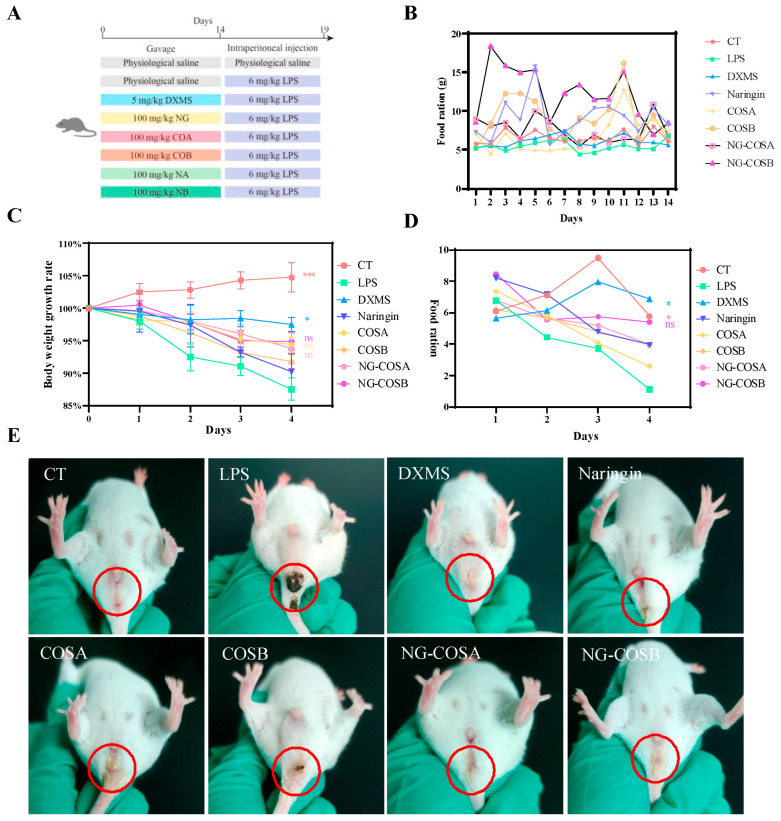
Animal design: the effect of the complex on mice with systemic inflammation. (**A**) Animal experimental design. (**B**) Average daily food intake during prophylactic gavage in mice. (**C**) Body weight changes in mice during LPS-induced inflammation. (**D**) Average daily food intake of mice during LPS-induced inflammation. (**E**) Body posture changes after LPS-induced inflammation in mice. The result was considered statistically significant when *p* < 0.05 between groups. * *p* < 0.05; *** *p* < 0.001; ns, not significant.

**Figure 2 foods-13-00576-f002:**
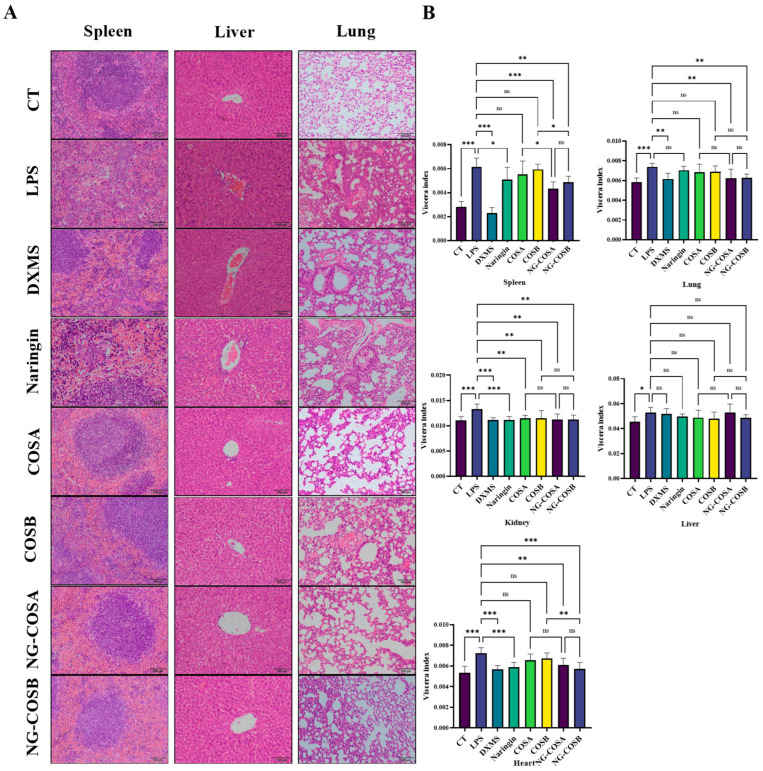
Histopathological analysis and the organ index of liver, lung, and spleen tissues in mice. (**A**) The representative images of H&E staining of liver, lung, and spleen tissues. (**B**) Effects of different experimental treatments on organ indexes in mice. The result was considered statistically significant when *p* < 0.05 between groups. * *p* < 0.05; ** *p* < 0.01; *** *p* < 0.001; ns, not significant.

**Figure 3 foods-13-00576-f003:**
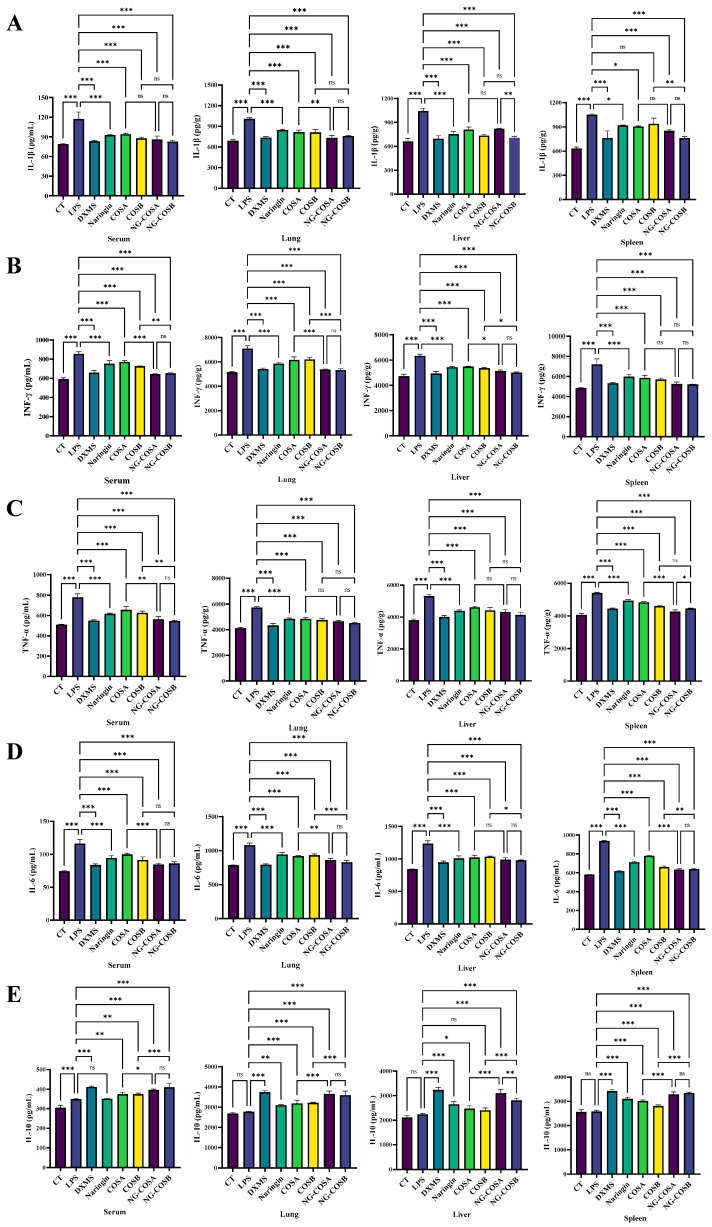
The effect of different experimental treatments on the serum and the tissue inflammation level is shown. (**A**) Content of IL-1β in serum and tissue. (**B**) Content of IFN-γ in serum and tissue. (**C**) Content of TNF-α in serum and tissue. (**D**) Content of IL-6 in serum and tissue. (**E**) Content of IL-10 in serum and tissue. The result was considered statistically significant when *p* < 0.05 between groups. * *p* < 0.05; ** *p* < 0.01; *** *p* < 0.001; ns, not significant.

**Figure 4 foods-13-00576-f004:**
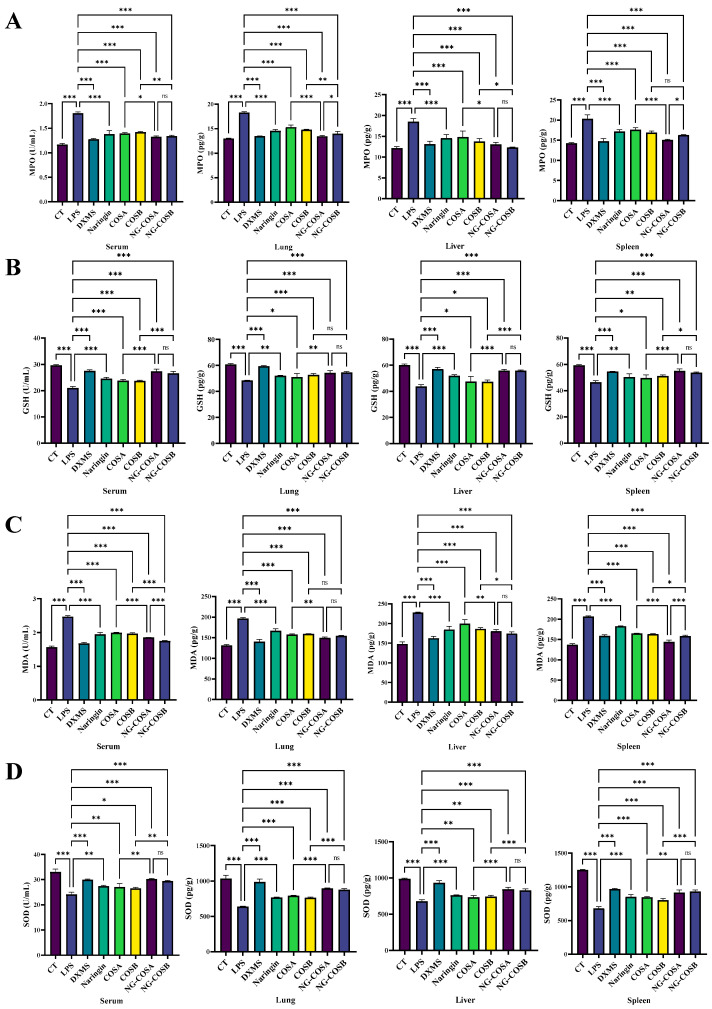
The effect of different experimental treatments on the serum and the tissues oxidative stress factor level is shown. (**A**) Content of MPO in serum and tissue. (**B**) Content of GSH in serum and tissue. (**C**) Content of MDA in serum and tissue. (**D**) Content of SOD in serum and tissue. The result was considered statistically significant when *p* < 0.05 between groups. * *p* < 0.05; ** *p* < 0.01; *** *p* < 0.001; ns, not significant.

**Figure 5 foods-13-00576-f005:**
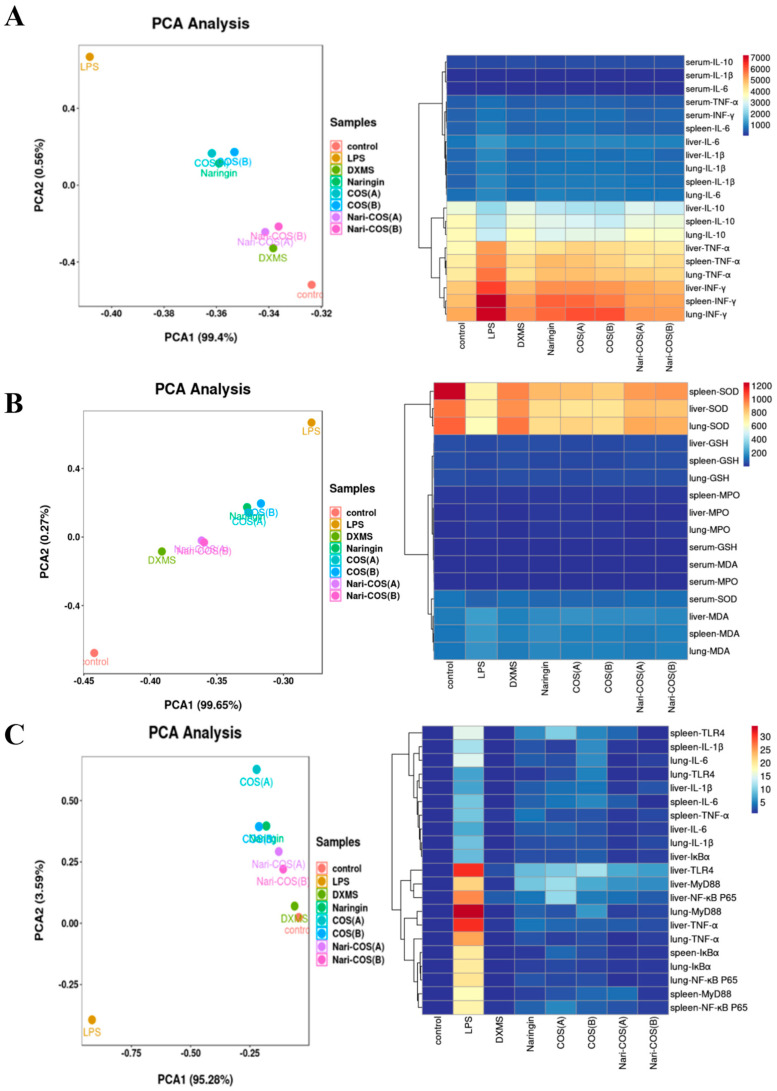
PCA analysis and visualization of the above results. The PCA analysis and heatmap visualization of (**A**) contents of inflammatory factors in serum and the tissues, (**B**) contents of oxidative stress factors in serum and the tissues, and (**C**) relative expression of inflammatory factor mRNA in the tissues.

**Table 1 foods-13-00576-t001:** Relative expression of inflammatory factor mRNA in mouse spleen.

Group	TLR4	IL-6	MyD88	IκBa	IL-1β	NF-κB p65	TNF-α
CT	1.0 ± 0.0 ^g^	1.0 ± 0.0 ^e^	1.0 ± 0.0 ^f^	1.0 ± 0.0 ^e^	1.0 ± 0.0 ^f^	1.0 ± 0.0 ^e^	1.0 ± 0.0 ^e^
LPS	30.7 ± 1.3 ^a^	7.6 ± 0.4 ^a^	21.7 ± 1.8 ^a^	8.7 ± 0.6 ^a^	7.2 ± 0.4 ^a^	26.1 ± 1.3 ^a^	30.7 ± 1.4 ^a^
DXMS	2.5 ± 0.4 ^f^	0.9 ± 0.4 ^e^	1.3 ± 0.1 ^f^	1.3 ± 0.2 ^de^	1.2 ± 0.1 ^ef^	3.1 ± 0.4 ^d^	1.3 ± 0.1 ^e^
Naringin	8.8 ± 0.7 ^cd^	3.1 ± 0.2 ^bc^	9.3 ± 0.5 ^c^	2.4 ± 0.2 ^c^	3.1 ± 0.4 ^d^	4.5 ± 0.2 ^c^	4.3 ± 0.2 ^b^
COSA	9.8 ± 1.4 ^c^	3.6 ± 0.5 ^b^	11.3 ± 0.8 ^b^	2.4 ± 0.5 ^c^	3.9 ± 0.1 ^c^	11.1 ± 0.7 ^b^	3.7 ± 0.3 ^bc^
COSB	11.7 ± 0.6 ^b^	2.6 ± 0.4 ^c^	7.3 ± 0.5 ^d^	3.2 ± 0.6 ^b^	4.4 ± 0.3 ^b^	4.6 ± 0.2 ^c^	3.3 ± 0.3 ^c^
NG-COSA	7.6 ± 0.4 ^de^	1.3 ± 0.1 ^de^	6.0 ± 0.1 ^e^	1.5 ± 0.4 ^de^	1.5 ± 0.3 ^e^	4.0 ± 0.1 ^cd^	3.1 ± 0.1 ^cd^
NG-COSB	6.7 ± 0.5 ^e^	1.6 ± 0.2 ^d^	5.6 ± 0.3 ^e^	1.9 ± 0.2 ^cd^	2.8 ± 0.1 ^d^	3.1 ± 0.2 ^d^	2.3 ± 0.2 ^d^

Note: Different letters in the same column indicate a significant difference, whereas the same letter indicates no significant difference, the same below.

**Table 2 foods-13-00576-t002:** Relative expression of inflammatory factor mRNA in mouse liver.

Group	TLR4	IL-6	MyD88	IκBa	IL-1β	NF-κB p65	TNF-α
CT	1.0 ± 0.0 ^g^	1.0 ± 0.0 ^e^	1.0 ± 0.0 ^f^	1.0 ± 0.0 ^e^	1.0 ± 0.0 ^f^	1.0 ± 0.0 ^e^	1.0 ± 0.0 ^e^
LPS	30.7 ± 1.3 ^a^	7.6 ± 0.4 ^a^	21.7 ± 1.8 ^a^	8.7 ± 0.6 ^a^	7.2 ± 0.4 ^a^	26.1 ± 1.3 ^a^	30.7 ± 1.4 ^a^
DXMS	2.5 ± 0.4 ^f^	0.9 ± 0.4 ^e^	1.3 ± 0.1 ^f^	1.3 ± 0.2 ^de^	1.2 ± 0.1 ^ef^	3.1 ± 0.4 ^d^	1.3 ± 0.1 ^e^
Naringin	8.8 ± 0.7 ^cd^	3.1 ± 0.2 ^bc^	9.3 ± 0.5 ^c^	2.4 ± 0.2 ^c^	3.1 ± 0.4 ^d^	4.5 ± 0.2 ^c^	4.3 ± 0.2 ^b^
COSA	9.8 ± 1.4 ^c^	3.6 ± 0.5 ^b^	11.3 ± 0.8 ^b^	2.4 ± 0.5 ^c^	3.9 ± 0.1 ^c^	11.1 ± 0.7 ^b^	3.7 ± 0.3 ^bc^
COSB	11.7 ± 0.6 ^b^	2.6 ± 0.4 ^c^	7.3 ± 0.5 ^d^	3.2 ± 0.6 ^b^	4.4 ± 0.3 ^b^	4.6 ± 0.2 ^c^	3.3 ± 0.3 ^c^
NG-COSA	7.6 ± 0.4 ^de^	1.3 ± 0.1 ^de^	6.0 ± 0.1 ^e^	1.5 ± 0.4 ^de^	1.5 ± 0.3 ^e^	4.0 ± 0.1 ^cd^	3.1 ± 0.1 ^cd^
NG-COSB	6.7 ± 0.5 ^e^	1.6 ± 0.2 ^d^	5.6 ± 0.3 ^e^	1.9 ± 0.2 ^cd^	2.8 ± 0.1 ^d^	3.1 ± 0.2 ^d^	2.3 ± 0.2 ^d^

Note: Different letters in the same column indicate a significant difference, whereas the same letter indicates no significant difference, the same below.

**Table 3 foods-13-00576-t003:** Relative expression of inflammatory factor mRNA in mouse lung.

Group	TLR4	IL-6	MyD88	IκBa	IL-1β	NF-κB p65	TNF-α
CT	1.0 ± 0.0 ^d^	1.0 ± 0.0 ^f^	1.0 ± 0.0 ^e^	1.0 ± 0.0 ^d^	1.0 ± 0.0 ^f^	1.0 ± 0.0 ^c^	1.0 ± 0.0 ^e^
LPS	6.9 ± 0.5 ^a^	14.7 ± 0.4 ^a^	34.0 ± 1.8 ^a^	19.7 ± 0.6 ^a^	9.1 ± 0.2 ^a^	20.5 ± 0.7 ^a^	24.9 ± 0.6 ^a^
DXMS	1.1 ± 0.1 ^d^	1.1 ± 0.1 ^f^	1.1 ± 0.0 ^e^	1.0 ± 0.2 ^d^	1.2 ± 0.1 ^ef^	1.0 ± 0.2 ^c^	1.0 ± 0.3 ^e^
Naringin	1.4 ± 0.1 ^d^	3.1 ± 0.2 ^c^	3.5 ± 0.4 ^c^	1.8 ± 0.1 ^bc^	2.3 ± 0.1 ^c^	2.8 ± 0.3 ^b^	2.8 ± 0.3 ^b^
COSA	2.3 ± 0.3 ^c^	2.5 ± 0.1 ^d^	2.6 ± 0.0 ^cd^	2.0 ± 0.2 ^b^	2.6 ± 0.1 ^b^	2.2 ± 0.3 ^b^	1.9 ± 0.1 ^c^
COSB	5.0 ± 0.3 ^b^	5.6 ± 0.1 ^b^	6.4 ± 0.2 ^b^	1.6 ± 0.2 ^bc^	2.6 ± 0.2 ^b^	2.5 ± 0.2 ^b^	2.5 ± 0.2 ^b^
NG-COSA	1.2 ± 0.1 ^d^	1.1 ± 0.0 ^f^	1.9 ± 0.1 ^de^	1.3 ± 0.0 ^cd^	1.3 ± 0.1 ^e^	1.0 ± 0.3 ^c^	1.2 ± 0.0 ^de^
NG-COSB	1.1 ± 0.1 ^d^	1.6 ± 0.3 ^e^	2.1 ± 0.2 ^de^	1.4 ± 0.0 ^cd^	2.0 ± 0.1 ^d^	1.4 ± 0.1 ^c^	1.6 ± 0.1 ^cd^

Note: Different letters in the same column indicate a significant difference, whereas the same letter indicates no significant difference, the same below.

## Data Availability

The original contributions presented in the study are included in the article/Appendix A; further inquiries can be directed to the corresponding authors.

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
