# Peer review of "Protective Effect of the Naringin–Chitooligosaccharide Complex on Lipopolysaccharide-Induced Systematic Inflammatory Response Syndrome Model in Mice"

_foods, 2024, doi:10.3390/foods13040576_

Round 1
Reviewer 1 Report
Comments and Suggestions for Authors
In this manuscript, the authors demonstrated to investigate the protective effect of Naringin-chitooligosaccharide Complex on LPS-induced SIRS model in mice. It was interesting experiment and knowledges, and PCA analysis is a good evaluation method that makes relationships very easy to understand.; however, there are some major concerns regarding data presentation, discussion and other issues that need to be considered. Important points should be reflected in the manuscript.
(Line 45)
Is there any possible effect of chitosan on allergic reactions?
(Lime 90)
The mice used in the experiment were a mixture of males and females, but if the mice were randomly selected, there may be a gender bias. Why did you perform the experiment with mixed sexes and what is the purpose of this experimental design? What do you predict about the effects of factors such as menstrual cycle and hormonal balance on results?
(Lime 96)
The weight of naringin in (4) is 100 mg, and are the weight ratio of naringin in (7) and (8) 100/6 mg?
(Lime 110)
There are some reports that the effectiveness of LPS varies depending on the type of bacteria, but what type of LPS bacteria were used in this experiment? Also, why was it set at 6 mg/kg?
(Lime 173)
What specific statistical methods were used to analyze the data?
(Lime 181)
Does the binding of COS to naringin affect the appetite stimulant effect due to changes in the oral environment caused by bitter and sweet tastes? Or is it due to the action exerted by each ingredient as they enter the organism?
(Lime 196)
How was the “mental” state of the mice measured and evaluated after LPS administration?
(Lime 218)
Is it possible that the increase in the liver organ index of DXMS, NG-COSA, and NG-COSB is due to the increase in food intake?
(Conclusions)
It has been reported that the glycoside portion of naringin is cleaved by intestinal bacteria, but is NG-COS absorbed and active without being cleaved? Although the amount of naringin in NG-COS was smaller than that of naringin alone, but it had a stronger effect, does this may be due to the increased absorbability of naringin?
Oligosaccharides serve as food for intestinal bacteria, and does COS play a role in activating intestinal bacteria and suppressing inflammation?
Reviewer 2 Report
Comments and Suggestions for Authors
The manuscript entitled “Protective Effect of Naringin-chitooligosaccharide Complex on
LPS-induced SIRS Model in Mice” is an interesting contribution to the devolepment of naringin-citooligosaccharide complex and their use against inflammatory process mediated by LPS. The introduction is well written, objective is clear, material and methods are reproducible, conclusions are supported by data, however, some details are needed.
Comments
Material and methods
Statistical analysis (Section 2.6)
Please add more details to the PCA analysis, i.e do you use a data treatment before use PCA?, Do you use several test to probe the obtained data before PCA analysis?
L175-176 Is not clear the affirmation “Principal component analysis 175 (PCA) was implemented to visualize the differentiation between groups” The main concern is the term “differentiation”, please clarify it, and,
Please add why the use of heat map?
The use of the animals was authorized by an ethical council or by the institutional authorities?
Results
L211 The authors claim: “Inflammation is typically characterized by increased endothelial tissue permeability”, however, they do not measure endothelial tissue permeability, please re-write.
General
Please, polish the figures
fig 2 is too big
P8 only have 2 lines
Fig 3 is too big
Fig 4 is too big
Tables 2 and 3 do not have information about superscript meaning (they do not have footnotes), only Table 1 showed footnote
P13 only have lines 332 to 343
Conclusions
The authors claim: “This study provides new ideas for the development and application of NG and NG-COS complex in the field of livestock and poultry feed and biomedicine. “ However, the authors do not show any experimental evidence to support the application of NG and NG-COS in the field of livestock and poultry feed and biomedicine, please delete or re-write.
Round 2
Reviewer 1 Report
Comments and Suggestions for Authors
The authors responded our suggestion carefully one by one and brush upped the manuscript.
Reviewer 2 Report
Comments and Suggestions for Authors
The revised manuscript entitled "Protective Effect of Naringin-chitooligosaccharide Complex on LPS-induced SIRS Model in Mice" Included all suggestion done by the reviewers, and answered all questions, thus, in my opinion is ready to be accept